# Infertility Treatment and Hypertension in Pregnancy: The Tohoku Medical Megabank Project Birth and Three-Generation Cohort Study

**Mami Ishikuro** [1,2,*], **Taku Obara** [1,2,3] , **Keiko Murakami** [1,2], **Fumihiko Ueno** [1,2], **Aoi Noda** [1,2,3], **Tomomi Onuma** [1,2], **Fumiko Matsuzaki** [1,2] , **Masahiro Kikuya** [1,4], **Zen Watanabe** [2,3], **Naomi Shiga** [2,3], **Masahito Tachibana** [2], **Noriyuki Iwama** [1,2,3], **Hirotaka Hamada** [2,3], **Masatoshi Saito** [2,3], **Junichi Sugawara** [1,2,3,5], **Hirohito Metoki** [1,6], **Nobuo Yaegashi** [1,2,3] and **Shinichi Kuriyama** [1,2,7]

1   Tohoku Medical Megabank Organization, Tohoku University, 2-1 Seiryo-machi, Aoba-ku, Sendai 980-8573, Japan
2   Graduate School of Medicine, Tohoku University, 2-1 Seiryo-machi, Aoba-ku, Sendai 980-8575, Japan
3   Tohoku University Hospital, 1-1 Seiryo-machi, Aoba-ku, Sendai 980-8574, Japan
4   School of Medicine, Teikyo University, 2-11-1 Kaga, Itabashi-ku, Tokyo 173-8605, Japan
5   Suzuki Memorial Hospital, 3-5-5, Satonomori, Iwanuma 989-2481, Japan
6   Faculty of Medicine, Tohoku Medical and Pharmaceutical University, 1-15-1, Fukumuro, Miyagino-ku, Sendai 983-8536, Japan
7   International Research Institute of Disaster Science, Tohoku University, 2-1 Seiryo-machi, Aoba-ku, Sendai 980-8573, Japan
*   Correspondence: m_ishikuro@med.tohoku.ac.jp

**Abstract:** Infertility treatment is a possible factor in hypertensive disorders of pregnancy (HDP). Identifying the characteristics of pregnant women who have undergone infertility treatment and have a potential risk for HDP is valuable for its prevention and treatment. Using data from 12,456 pregnant Japanese women from the Tohoku Medical Megabank Project Birth and Three-Generation Cohort Study, the association between infertility treatment and HDP was analyzed. A multiple logistic regression model showed an association between infertility treatment and HDP (odds ratio, 1.34; 95% confidence interval, 1.05–1.72). In vitro fertilization/intracytoplasmic sperm injection were also associated with HDP. Moreover, these associations were observed even among women who were not overweight and did not smoke. The application of infertility treatment should be carefully considered, even among women with low modifiable risk factors.

**Keywords:** assisted conception; assisted reproductive technology; treatment for infertility; hypertensive disorders of pregnancy; cohort study

## 1. Introduction

Hypertensive disorders of pregnancy (HDP) are a common obstetric complication. The incidence of HDP increased by approximately 11% from 1990 to 2019 [1]. Although high-income countries have a lower incidence of HDP than middle- or low-income countries [1], HDP is a risk factor for maternal death and is the cause of 14% of maternal deaths in Japan [2]. HDP can also lead to chronic hypertension [3]. Therefore, it is important to prevent HDP.

Treatment is administered to couples experiencing infertility. It is reported that infertility treatment is a possible risk factor for HDP [4]. In particular, assisted reproductive technology (ART) has been associated with HDP in previous studies [5,6]. Since the first successful case of in vitro fertilization (IVF) in 1978, ART applications have increased globally, and more than seven million children have been conceived using ART [7]. According to the ART registry of the Japan Society of Obstetrics and Gynecology [8], 5.4–8.2% of children born per year between 2010 and 2019 were conceived using ART [9]. In 2016, 447,763 cycles

of ART were performed in Japan, which is approximately 2.3 times higher than that in the United States [10]. As part of ART, IVF and intracytoplasmic sperm injection (ICSI) require fertilization of the oocytes outside the woman's body. A recent meta-analysis showed that both IVF and ICSI have higher odds ratios for HDP than spontaneous pregnancy [11]. A nationwide birth cohort study in Japan that included pregnant women between 2011 and 2014 also reported similar results [12]. However, the background of women who develop HDP is not only infertility treatment but also a variety of other risk factors, such as age, parity, a history of HDP, multiple pregnancies, overweight/obesity, smoking, and so on. Identifying the characteristics of pregnant women who have undergone infertility treatment and have a potential risk for HDP is valuable for preventing and treating HDP.

The objective of this study was to investigate the association between infertility treatment and HDP in pregnant Japanese women stratified by other modifiable factors for HDP.

## 2. Materials and Methods

### 2.1. Design

Data from the Tohoku Medical Megabank Project Birth and Three-Generation Cohort Study (TMM BirThree Cohort Study) [13–18] were used for the analysis. The TMM BirThree Cohort Study was launched in Miyagi Prefecture, Japan, and recruited pregnant women and their family members between 2013 and 2017 [13–16]. The purpose of the study was to investigate the effects of the Great East Japan Earthquake on people's health, identify their need for healthcare services, and establish precision medicine/healthcare to provide better care for people living in the disaster area and all over Japan. In the brief introduction of the TMM BirThree Cohort Study, pregnant women were invited to participate in the study at obstetric hospitals or clinics when they booked their deliveries. Some pregnant women were also recruited at their own research facilities, which are established for voluntary-based recruitment and health assessment. Genome medical research coordinators explained the TMM BirThree Cohort Study to pregnant women and obtained their signed informed consent. After pregnant women participated in the TMM BirThree Cohort Study, their family members were also invited to participate. Finally, more than 22,000 mother-child pairs, about 9000 child siblings, 16,000 fathers and grandparents, and 1400 other family members [13] participated in the TMM BirThree Cohort Study, totaling over 70,000 family members. The mean maternal age was 31.4 years old. The use of ART in all pregnant women was about 6.6%, which was comparable to nationwide Japanese data [15].

### 2.2. Participants

The participants of the TMM BirThree Cohort Study were 22,493 pregnant women. Participants who withdrew their consent (*n* = 575), had a history of hypertension or missing information about hypertension history, used antihypertensive drugs before pregnancy, had missing information or unmatched information regarding infertility treatment in the self-reported questionnaire or medical records, or had missing information for covariates (*n* = 9462) were excluded (Figure 1).

### 2.3. Measurements

Data on the different types of infertility treatments was collected using questionnaires and medical records. The types of infertility treatments included ovulation induction, artificial insemination, IVF, and ICSI. We only considered participants whose responses to the questionnaires matched their medical records.

Data on multiple pregnancies was obtained from the medical records. Maternal age at conception was calculated based on their date of birth and the date of childbirth as provided in the medical records. Accordingly, the participants were categorized into two groups: <35 years old and ≥35 years old. Based on the information on parity obtained from the medical records, the participants were categorized into three groups: nulliparous women, multiparous women without a history of HDP, and multiparous women with a history of HDP. Data on paternal or maternal history of hypertension or history of HDP

were obtained from the postpartum questionnaire, based on which the participants were categorized into those with or without a family history of hypertension. Pre-pregnancy body mass index (BMI) was calculated based on the data on height and weight provided in the first trimester questionnaire, and based on this, the participants were categorized into two groups: $<25 \text{ kg/m}^2$ and $\geq 25 \text{ kg/m}^2$. Based on the information regarding smoking status obtained from the first trimester questionnaire, the participants were categorized into two groups: the "never" or "quit" group and the "continued in early pregnancy" group. Pre-pregnancy BMI and smoking were targeted as modifiable risk factors for HDP.

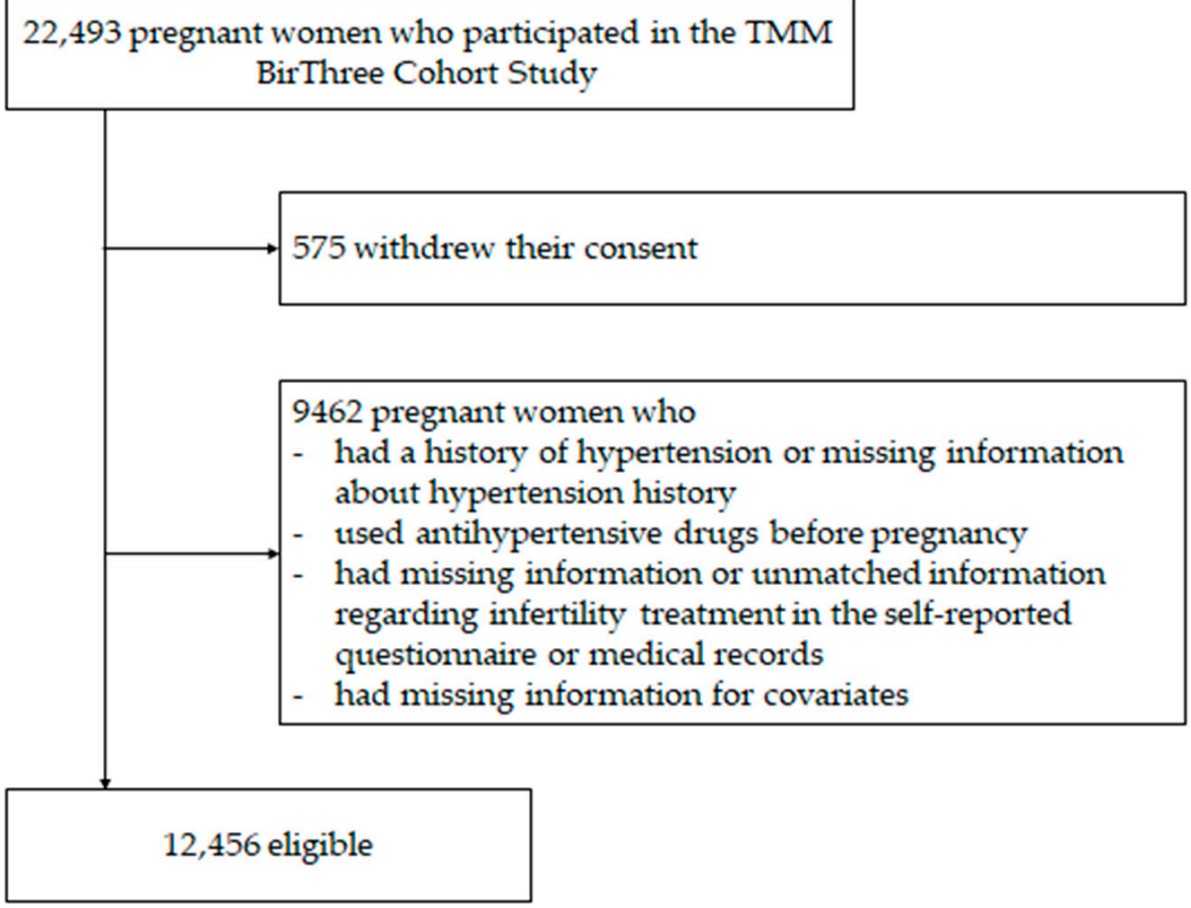

**Figure 1.** Participant selection diagram.

### 2.4. Outcome

HDP was defined using an original algorithm of the TMM BirThree Cohort Study in accordance with the criteria of the American College of Obstetricians and Gynecologists [19–21]. Blood pressure (BP) levels and urological test results were collected throughout pregnancy from the medical records provided by the Genome Medical Research Coordinators. Systolic BP $\geq 140$ mmHg or diastolic BP $\geq 90$ mmHg, even once during pregnancy, was categorized as HDP. The classification was validated by a clinician for 0.2% of the participants. Women who used antihypertensive drugs during pregnancy were categorized as having HDP.

### 2.5. Statistical Analysis

The characteristics of pregnant women who underwent infertility treatment and those who conceived without infertility treatment were compared using the chi-square test. The characteristics between them were also compared between single and multiple pregnancies. The logistic regression model was adjusted for maternal age at conception, pre-pregnancy

BMI, smoking, family history of hypertension, and multiple pregnancies. Variables for adjustment were selected from the possible factors for HDP [22,23] and confounders based on previous studies [24,25]. The association between infertility treatment and HDP was also investigated for single and multiple pregnancies. In addition, exposure was limited to IVF and ICSI, and a logistic regression analysis was performed to investigate their association with HDP.

Sub-group analyses stratified by pre-pregnancy BMI and smoking were also performed to investigate the association between infertility treatment or IVF/ICSI and HDP. In addition, we divided women who underwent IVF/ICSI into those who underwent fresh and frozen embryo transfers and described the prevalence of HDP in each group using the chi-square test. Information was only available from the medical records; therefore, we did not perform further analysis as the main outcome. All statistical analyses were performed using SAS software (version 9.4; SAS Institute Inc., Cary, NC, USA).

## 3. Results

### 3.1. Maternal Characteristics

A total of 12,456 pregnant women were included in this study. The prevalence of HDP was 14.5% in women who underwent infertility treatment, compared to only 9.4% in women who conceived naturally. There was a higher percentage of women aged ≥35 years at conception (51.8%) and nulliparity among women treated for infertility (71.2%) than among those who did not undergo infertility treatment (24.5 and 45.0%, respectively). The family history of hypertension was 39.3% in women who underwent infertility treatment and 33.9% in women who conceived naturally. The percentages of overweight women among those with infertility (13.0%) and those who had spontaneous pregnancies (10.6%) were comparable. The proportion of smokers was lower among those who underwent infertility treatment than among those who did not undergo treatment (Table 1).

In this study, 153 women had multiple pregnancies. The characteristics of women who did and did not undergo infertility treatment were similar to those of women who had a single pregnancy, except regarding the family history of hypertension (Table 1).

### 3.2. Infertility Treatment and HDP

A multiple logistic regression model showed an association between infertility treatment and HDP (odds ratio (OR), 1.34; (95% confidence interval (CI), 1.05–1.72). The OR for single pregnancy was 1.32 (95% CI, 1.02–1.70), whereas it was 3.21 (95% CI, 0.97–10.61) for multiple pregnancy (Table 2). For IVF/ICSI, the OR in the multiple logistic regression model was 1.65 (95% CI, 1.20–2.29). The OR for single pregnancy was 1.61 (95% CI, 1.16–2.25), whereas it was 4.31 (95% CI, 0.74–25.16) for multiple pregnancy (Table 3).

### 3.3. Subgroup Analysis by Overweight and Smoking

In the subgroup analysis of women who were overweight or not overweight, infertility treatment was associated with HDP in the latter. This association was also observed when infertility treatment was restricted to IVF/ICSI. Furthermore, an association between infertility treatment and HDP was observed in women who did not smoke (Table 4).

**Table 1.** Characteristics of pregnant women who did and did not undergo infertility treatment in the total and single/multiple pregnancy groups.

| | Total (n = 12,456) | | | | | Single Pregnancy (n = 12,303) | | | | | Multiple Pregnancy (n = 153) | | | | |
|---|---|---|---|---|---|---|---|---|---|---|---|---|---|---|---|
| | No Infertility Treatment | | Underwent Infertility Treatment | | p Value | No Infertility Treatment | | Underwent Infertility Treatment | | p Value | No Infertility Treatment | | Underwent Infertility Treatment | | p Value |
| | n | % | n | % | | n | % | n | % | | n | % | n | % | |
| Age at conception | | | | | <0.0001 | | | | | <0.0001 | | | | | <0.0001 |
| <35 years | 8956 | 75.5 | 288 | 48.2 | | 8861 | 75.5 | 272 | 48.0 | | 95 | 77.9 | 16 | 51.6 | |
| ≥35 years | 2902 | 24.5 | 310 | 51.8 | | 2875 | 24.5 | 295 | 52.0 | | 27 | 22.1 | 15 | 48.4 | |
| Pre-pregnancy BMI | | | | | 0.06 | | | | | 0.1 | | | | | 0.08 |
| <25 kg/m$^2$ | 10,597 | 89.4 | 520 | 87.0 | | 10,484 | 89.3 | 495 | 87.3 | | 113 | 92.6 | 25 | 80.7 | |
| ≥25 kg/m$^2$ | 1261 | 10.6 | 78 | 13.0 | | 1252 | 10.7 | 72 | 12.7 | | 9 | 7.4 | 6 | 19.4 | |
| Parity | | | | | <0.0001 | | | | | <0.0001 | | | | | 0.3 |
| Nulliparous women | 5338 | 45.0 | 426 | 71.2 | | 5277 | 45.0 | 406 | 71.6 | | 61 | 50.0 | 20 | 64.5 | |
| Multiparous women with a history of HDP | 267 | 2.3 | 7 | 1.2 | | 262 | 2.2 | 7 | 1.2 | | 5 | 4.1 | 0 | 0 | |
| Multiparous women without a history of HDP | 6253 | 52.7 | 165 | 27.6 | | 6197 | 52.8 | 154 | 27.2 | | 56 | 45.9 | 11 | 35.5 | |
| Family history of hypertension | | | | | 0.007 | | | | | 0.002 | | | | | 0.2 |
| No | 7837 | 66.1 | 363 | 60.7 | | 7758 | 66.1 | 339 | 59.8 | | 79 | 64.8 | 24 | 77.4 | |
| Yes | 4021 | 33.9 | 235 | 39.3 | | 3978 | 33.9 | 228 | 40.2 | | 43 | 35.3 | 7 | 22.6 | |
| Smoking | | | | | <0.0001 | | | | | <0.0001 | | | | | 0.02 |
| Never/quit | 10,606 | 89.4 | 582 | 97.3 | | 10,501 | 89.5 | 551 | 97.2 | | 105 | 86.1 | 31 | 100 | |
| Continued in early pregnancy | 1252 | 10.6 | 16 | 2.7 | | 1235 | 10.5 | 16 | 2.8 | | 17 | 13.9 | 0 | 0 | |

BMI; body mass index, HDP; hypertensive disorders of pregnancy.

**Table 2.** Association between infertility treatment and HDP in total and single/multiple pregnancy.

| | Number of HDP/Pregnant Women Who Underwent Infertility Treatment, HDP/Pregnant Women Who Did Not Undergo Infertility Treatment | Crude OR | 95% CI | Adjusted OR | 95% CI |
|---|---|---|---|---|---|
| Total | 87/598, 1120/11,858 | 1.63 | 1.29–2.07 | 1.34 | 1.05–1.72 |
| Single pregnancy | 79/567, 1103/11736 | 1.56 | 1.22–2.00 | 1.32 | 1.02–1.70 |
| Multiple pregnancy | 8/31, 17/122 | 2.15 | 0.83–5.58 | 3.21 | 0.97–10.61 |

HDP, hypertensive disorders of pregnancy; OR, odds ratio; CI, confidence interval.

**Table 3.** Association between IVF/ICSI and HDP in total and single/multiple pregnancy.

| | Number of HDP/Pregnant Women Who Underwent IVF/ICSI, HDP/Pregnant Women Who Did Not Undergo Infertility Treatment | Crude OR | 95% CI | Adjusted OR | 95% CI |
|---|---|---|---|---|---|
| Total | 50/307, 1120/11,858 | 1.87 | 1.37–2.54 | 1.65 | 1.20–2.29 |
| Single pregnancy | 47/298, 1103/11,736 | 1.81 | 1.31–2.48 | 1.61 | 1.16–2.25 |
| Multiple pregnancy | 3/9, 17/122 | 3.09 | 0.71–13.53 | 4.31 | 0.74–25.16 |

IVF, in vitro fertilization; ICSI, intracytoplasmic sperm injection; HDP, hypertensive disorders of pregnancy; OR, odds ratio; CI, confidence interval.

*3.4. Proportion of HDP in Fresh or Frozen Embryo Transfer*

Of the 307 pregnant women who underwent IVF/ICSI, 247 had data on fresh or frozen embryo transfer. The proportion of pregnant women who developed HDP was 9.5% in the fresh embryo transfer group, whereas 19.0% of pregnant women in the frozen embryo transfer group developed HDP ($p = 0.1$).

**Table 4.** Analyses of subgroups stratified by pre-pregnancy BMI and smoking history.

| Infertility treatment | | | |
|---|---|---|---|
| Pre-pregnancy BMI | Adjusted OR | 95% CI | *p* value for interaction |
| <25 kg/m$^2$ | 1.36 | 1.03–1.80 | 0.5 |
| ≥25 kg/m$^2$ | 1.25 | 0.74–2.11 | |
| Smoking | Adjusted OR | 95% CI | *p* value for interaction |
| Never/quit | 1.36 | 1.06–1.75 | 0.7 |
| Continued in early pregnancy | 1.79 | 0.49–6.58 | |
| IVF/ICSI | | | |
| Pre-pregnancy BMI | Adjusted OR | 95% CI | *p* value for interaction |
| <25 kg/m$^2$ | 1.78 | 1.27–2.51 | 0.12 |
| ≥25 kg/m$^2$ | 0.90 | 0.35–2.29 | |
| Smoking | Adjusted OR | 95% CI | *p* value for interaction |
| Never/quit | 1.72 | 1.24–2.38 | NA |
| Continued in early pregnancy | <0.001 | <0.001–>999.9 | |

IVF, in vitro fertilization; ICSI, intracytoplasmic sperm injection; HDP, hypertensive disorders of pregnancy; OR, odds ratio; CI, confidence interval.

## 4. Discussion

The present study found that infertility treatment is associated with a higher prevalence of HDP. IVF/ICSI has also been associated with a higher prevalence of HDP. A nationwide birth cohort study in Japan, which reported the association between ART and HDP, used women's self-reports for the information on ART [12]. This study obtained information on infertility treatments, including IVF/ICSI, from self-reported questionnaires completed by women and through medical records in the TMM BirThree Cohort Study; therefore, it adds reliable information to the previous study. Furthermore, this study reported a more recent situation in Japan than the previous study. Furthermore, this study found that both infertility treatment and IVF/ICSI are associated with the prevalence of HDP, even among women who are not overweight or smokers.

This study investigated ovulation induction, artificial insemination, IVF, and ICSI as potential infertility treatments. Dobrosavljevic et al. reported that ovarian hyperstimulation syndrome, which is often caused by ovulation stimulation, was a possible factor for gestational hypertension among women who underwent IVF/ICSI; however, they also mentioned that ovarian hyperstimulation syndrome caused vascular permeability, hemoconcentration, hemodynamic instability, and influenced placentation; therefore, morbidity rather than ovulation stimulation might be a trigger for HDP [26]. Another study that reported an association between IVF and a higher prevalence of hypertension during pregnancy among patients with ovulation disorders also supported the idea that ovulation induction might be indirectly associated with HDP [27]. Meanwhile, artificial insemination is used more for male infertility than female infertility [28]. Older women were more likely to have older partners; therefore, many cases may have a combined type of subfertility. However, the TMM BirThree Cohort Study did not collect any information on male infertility. Further studies focusing on the differences between male and female infertility factors are warranted.

Similar to previous systematic reviews and meta-analyses [5,6,11], IVF and ICSI were associated with an increased risk of HDP in this study. Furthermore, several studies have pointed out that frozen embryo transfer may increase the risk of HDP [29,30]. We could not investigate the association of frozen embryo transfer with the increased risk of HDP due to an insufficient number of participants and the availability of data sources; however, the proportion of HDP in the frozen embryo transfer group was higher than that in the fresh embryo transfer group. The mechanism underlying the possible risk of frozen embryo transfer in HDP remains unclear. However, a recent study focusing on the hormonal environment suggested that the absence of the corpus luteum results in a lack of circulating relaxin, thereby attenuating an increase in central arterial compliance and increasing the

incidence of preeclampsia [31]. In addition, Sites et al. reported that removing some cells for genetic testing of cryopreserved embryos before implantation was not associated with an increased risk of different maternal outcomes [32]. A recent study found that frozen embryo transfer during the programmed cycle was associated with an increased risk of HDP compared to the natural cycle [33]. Further studies are essential to assess the consequences of the risk of frozen embryo transfer regarding abnormal maternal outcomes.

The application of infertility treatment is still considered to increase a potential risk of HDP in women, as indicated in a study involving IVF [24]. As a possible HDP prevention strategy, Liu et al. reported that a 5% reduction in BMI before IVF may be associated with a decreased risk of HDP for women who are overweight, and a 10% reduction in BMI before IVF may be associated with a decreased risk of HDP for women who are obese [34]. However, in this study, the association between infertility treatment and HDP was confirmed not only in pregnant women who were overweight but also in those who were not overweight. Regardless of BMI, IVF/ICSI should be considered a potential risk factor for HDP when it applies to women. Whether smoking is a factor that increases or decreases the risk of HDP is still debatable [35]. However, smoking may have the potential to increase the risk of HDP in pregnant Japanese women [23]. In pregnant Japanese women, secondhand smoke is also associated with an increased risk of HDP. Tanaka et al. analyzed data from about 100,000 pregnant women and found that the relative risk of women who were exposed to secondhand smoke four or more days a week was 1.18, compared to women who rarely had secondhand smoke exposure [36]. Furthermore, the population attributable fraction for the risk of HDP due to secondhand smoke was 3.8%. Therefore, smoking may need to be considered when preventing HDP in pregnant Japanese women. In this study, the proportion of pregnant women who smoked among those who underwent infertility treatment was lower than that among those who did not. This is speculated to be because the women who underwent infertility treatment have a higher socio-economic status and pay more attention to their health. Nevertheless, the association between infertility treatment and HDP was still observed in women who never smoked or quit smoking. Monitoring BP after conception is necessary, even among low-risk modifiable factors for HDP. In addition to the findings, if infertility treatment affects the onset or severity of HDP.

A report regarding the time trend of HDP based on data from the national health registries of Denmark, Finland, and Sweden between 1990 and 2015 demonstrated that the risk of HDP was higher in pregnancies that used ART than in spontaneous pregnancies [37]. Furthermore, ART in multiple pregnancies was associated with a greater risk of HDP, despite a decrease in the rate of multiple pregnancies throughout the study period. In Japan, owing to the declining birth rate, infertility treatment such as IVF is covered by the national health insurance system. The present study, which included a more recent population than a previous Japanese study [12], found an association between IVF/ICSI and a higher prevalence of HDP. Considering these studies, clinical facilities should carefully assess the application of infertility treatment in women from the point of view of a possible prognosis after conception. HDP is also a risk factor for a higher prevalence of hypertension later in life [3], especially in younger women [38]. In their 30s, women who have experienced HDP have an OR of hypertension of 3.63 compared to women without a history of HDP. If hypertension persists into later life, it could become a significant burden for women. ART is associated with a potential risk of preterm birth. In particular, HDP mediates 32.6 and 25.9% of the association between frozen embryo transfer and early preterm birth and late preterm birth, respectively [39]. HDP itself is also a well-known factor for preterm birth and low birthweight. According to a meta-analysis, ORs of preterm birth and low birthweight were 4.20 and 5.02, respectively, among women with HDP compared to women without HDP [40]. Among women who underwent ART, HDP was also positively associated with preterm birth and low birthweight in both singleton and multiple pregnancies [41]. The ORs of preterm birth and low birthweight among single pregnancies were higher than those among multiple pregnancies. Not only HDP but also a higher BP trajectory during

pregnancy may affect child birthweight [42]. Iwama et al. revealed that pregnant women in the high J-curve group, compared to those in the low J-curve group, delivered children with lower birthweight. Perinatal outcomes should also be considered in the maternal HDP. Iwama et al. used home BP values for the analysis [42]. The target threshold for HDP is defined by the International Society for the Study of Hypertension in Pregnancy [43]. In order to monitor BP after conception, home BP monitoring in addition to clinic BP monitoring may be applicable.

The strength of the present study is that it considered self-reports and medical records for the analyses. Self-reports may have recall bias. However, sometimes the facility handling IVF/ICSI is different from the one where the patient is sent for delivery; therefore, complete information on IVF/ICSI was not available from the medical records. This study has some limitations. First, no information on the drugs used for ovulation induction was included in the study; therefore, the effect of drugs on the development of HDP could not be analyzed. Second, this study used the original algorithm of the TMM BirThree Cohort Study, whose criteria might have overestimated the prevalence of HDP, although they have been validated by a clinician. However, the prevalence of HDP in the participants in the present study was 10.0%, which is similar to that reported in other studies [20,44]. Third, some pre-pregnancy potential confounders or factors for HDP, such as physical activity [45] were not collected; therefore, further study concerning pre-pregnancy potential confounders is necessary.

## 5. Conclusions

The application of infertility treatment should be carefully considered, even among women with low modifiable risk factors.

**Author Contributions:** Conceptualization and methodology, M.I. and T.O. (Taku Obara); software, M.I.; validation, K.M., F.U. and A.N.; formal analysis, M.I.; investigation and resources, M.I., T.O. (Taku Obara), K.M., F.U., A.N., T.O. (Tomomi Obara), F.M., M.K., Z.W., N.S., M.T., N.I., H.H., M.S., J.S., H.M., N.Y. and S.K.; data curation, T.O. (Taku Obara), K.M., F.U., A.N., T.O. (Tomomi Obara) and F.M.; writing—original draft preparation, M.I.; writing—review and editing, T.O. (Taku Obara), K.M., F.U., A.N., T.O. (Tomomi Obara), F.M., M.K., Z.W., N.S., M.T., N.I., H.H., M.S., J.S., H.M., N.Y. and S.K.; visualization, M.I.; supervision, T.O. (Taku Obara) and S.K.; project administration, N.Y. and S.K.; funding acquisition, N.Y. and S.K. All authors have read and agreed to the published version of the manuscript.

**Funding:** This study was supported by the Japan Agency for Medical Research and Development (AMED, Japan) (grant numbers, JP17km0105001, JP21tm0124005, and JP19gk0110039).

**Institutional Review Board Statement:** The study was conducted in accordance with the Declaration of Helsinki, and the protocol of the TMM BirThree Cohort Study was approved by the Tohoku Medical Megabank Organization Internal Review Board (no. 2013-1-103-1, approved on 27 May 2013).

**Informed Consent Statement:** Written informed consent was obtained from all participants involved in the study.

**Data Availability Statement:** The data supporting the findings of this study are not publicly available because they contain information that could compromise research participant consent. All inquiries regarding access to the data should be sent to the TMM (dist@megabank.tohoku.ac.jp).

**Acknowledgments:** We appreciate the participants, medical staff at hospitals and clinics, municipalities, schoolteachers, and other stakeholders who supported the TMM BirThree Cohort Study. A full list of members of the Tohoku Medical Megabank Organization is available at https://www.megabank.tohoku.ac.jp/english/a220901/ (accessed on 17 March 2023) for ToMMo.

**Conflicts of Interest:** The authors declare no conflict of interest.

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
