# Peer review of "Infertility Treatment and Hypertension in Pregnancy: The Tohoku Medical Megabank Project Birth and Three-Generation Cohort Study"

_2673-3897, doi:10.3390/reprodmed4020010_

Round 1
Reviewer 1 Report
This is an interesting paper discussing hypertension disorders in pregnancy obtained with IVF
the manuscript is well written and discussed also the last news about IVF and hypertension
Author Response
Thank you very much for reviewing our manuscript. We revised the manuscript according to other reviewers. We submit the revised manuscript.
Reviewer 2 Report
The current study by Ishikuro et al confirms previous reports on an association between subfertility treatment and hypertension in pregnancy. The study offers few if any additional insights.
I have no major questions or comments, just a few minor ones:
- Discussion, page 6, line 183: male or female infertility. It probably does not matter much: older women tend to have older partners, with a higher risk of poor sperm quality. In many cases, there is a combined type of subfertility.
- Discussion, page 6, line 208. Smoking has been shown to protect against pre-eclampsia, which may be worth citing.
A lower smoking rate may reflect a higher socio-economic status, which correlates highly with health-promoting behaviour. Couples with higher SES tend to be older when trying to conceive.
Author Response
Thank you very much for giving us the opportunity to revise our manuscript. In response to your suggestions, we minorly revised the manuscript.
(Comment 1)
- Discussion, page 6, line 183: male or female infertility. It probably does not matter much: older women tend to have older partners, with a higher risk of poor sperm quality. In many cases, there is a combined type of subfertility.
(Answer 1)
Thank you very much for your advice. We added a sentence in the manuscript as below.
P7, Line 195-199
Older women were more likely to have older partners; therefore, many cases may have combined type of subfertility. However, the TMM BirThree Cohort Study did not collect any information on male infertility. Further studies focusing on the differences between male and female infertility factors are warranted.
(Comment 2)
- Discussion, page 6, line 208. Smoking has been shown to protect against pre-eclampsia, which may be worth citing.
(Answer 2)
We added a reference as below, and the rest of references were reordered.
Page 7, Line 222-Page 8, Line 223
Whether smoking is a factor that increases or decreases the risk of HDP is still debatable [32]; however, smoking may have the potential to increase the risk of HDP in pregnant Japanese women [21].
(Comment 3)
A lower smoking rate may reflect a higher socio-economic status, which correlates highly with health-promoting behaviour. Couples with higher SES tend to be older when trying to conceive.
(Answer 3)
We appreciate for the comment. We revised the following sentences based on your comment.
Page 8, Line 225-227
This is speculated to be because the women who underwent infertility are higher socio-economic status, and who wish to become pregnant pay more attention to their health.
Reviewer 3 Report
Title: Infertility treatment and hypertension in pregnancy: the Tohoku Medical Megabank Project Birth and Three-Generation Cohort Study
In this study, authors investigated the association between infertility treatment and risk of hypertension during pregnancy. Indeed it is a well written, concise, and interesting study, however, I only have few minor consideration.
1. Inclusion/exclusion, authors adequately described about all criterion but I am wondering about the physical activity that might be an important confounder for HDP.
2. Similarly, I found no information about the pregnancy trimester and its association between infertility treatment and HDP.
3. It is better to include few lines in the introduction section about other factors that might be associated with HDP.
Author Response
We appreciate for giving us the opportunity to revise our manuscript. We hope that we made a correction following your comments.
(Comment 1)
Inclusion/exclusion, authors adequately described about all criterion but I am wondering about the physical activity that might be an important confounder for HDP.
(Answer 1)
Thank you very much for your advice. Pre-pregnancy physical activity may be related to a lower risk of gestational hypertension; however, we did not collect information about it. We added it in the discussion section.
Page 8, Line 256-258
Third, some pre-pregnancy potential confounder or factors for HDP such as physical activity [36] was not collected; therefore, further study concerning pre-pregnancy potential confounders is necessary.
(Comment 2)
Similarly, I found no information about the pregnancy trimester and its association between infertility treatment and HDP.
(Answer 2)
As you suggested, to investigate whether infertility treatment affects early or late onset of HDP is important, as well as severity of HDP. We added the sentence in discussion section as below.
Page 8, Line 229-231
In addition to the findings, if infertility treatment affects onset or severity of HDP is another issue to be clarified.
(Comment 3)
It is better to include few lines in the introduction section about other factors that might be associated with HDP.
(Answer 3)
Thank you very much for your valuable comment. We added the other factors that might be associated with HDP.
Page 2, Line 52-55
However, the background of women who develop HDP is not only infertility treatment but also a variety of other risk factors, such as age, parity, history of HDP, multiple pregnancy, overweight/obesity, smoking and so on.
Reviewer 4 Report
In this study, the authors aimed to analyze the association between infertility treatment and hypertensive disorders of pregnancy by using the data from TMM BirThree cohort study. The results indicated that IVF/ICSI treatments were associated with hypertensive disorders of pregnancy, independently of obesity and smoking. This set of data is of interest, providing some epidemiologic data for ART treatments.
It is better to use a schematic diagram to illustrate the inclusion and exclusion criteria and cases in this analysis.
The result part should be subdivided into several subtitles and the elaboration should be more specified.
The proportion of advanced age women with ART treatments was significantly higher than women without ART treatments. Advanced age is an identified risk factor for hypertensive disorder of pregnancy as well as other pregnancy complications, while the association between ART and HDP in advanced age women was not analyzed. What was the incidence of HDP in advanced age women with or without ART treatments?
The authors indicated that IVF/ICSI treatments were associated with increased incidence of HDP, however, the manuscript used ART treatments or infertility treatment, IVF/ICSI was part of ART treatments, it can not take the place of ART treatments, embryo cryopreservation, frozen embryo transfers, preimplantation genetic testing, artificial insemination are all belongs to ART treatments. The authors need to clarify the inclusion criteria in women with ART treatments, did they all received IVF/ICSI, fresh embryo transfers or frozen embryo transfers?
Author Response
Thank you very much for giving us an opportunity to improve our manuscript. We revised manuscript based on your advice and answered your questions.
(Comment 1)
It is better to use a schematic diagram to illustrate the inclusion and exclusion criteria and cases in this analysis.
(Answer 1)
Thank you very much for your recommendation. We added the diagram as Figure 1.
(Comment 2)
The result part should be subdivided into several subtitles and the elaboration should be more specified.
(Answer 2)
Based on your advice, we subdivided the result part.
(Comment 3)
The proportion of advanced age women with ART treatments was significantly higher than women without ART treatments. Advanced age is an identified risk factor for hypertensive disorder of pregnancy as well as other pregnancy complications, while the association between ART and HDP in advanced age women was not analyzed. What was the incidence of HDP in advanced age women with or without ART treatments?
(Answer 3)
Thank you very much for your valuable comment. Indeed, older women have higher risk for HDP. In the analysis, the logistic regression model was adjusted for age 35 years or more, and the crude and adjusted odds ratio (95% confidence interval) was 1.13 (0.98-1.29). We would have liked to investigate the association between infertility treatment and HDP in modifiable factors. Therefore, we did subgroup analysis without age stratification.
(Comment 4)
The authors indicated that IVF/ICSI treatments were associated with increased incidence of HDP, however, the manuscript used ART treatments or infertility treatment, IVF/ICSI was part of ART treatments, it can not take the place of ART treatments, embryo cryopreservation, frozen embryo transfers, preimplantation genetic testing, artificial insemination are all belongs to ART treatments. The authors need to clarify the inclusion criteria in women with ART treatments, did they all received IVF/ICSI, fresh embryo transfers or frozen embryo transfers?
(Answer 4)
We appreciate for your advice. We clarified that we analyzed the association between IVF/ICSI and HDP instead of using the word “ART” in Table 3 and 4, and discussion section.